# Comparison of the Results of Cardiopulmonary Exercise Testing between Healthy Peers and Pediatric Patients with Different Echocardiographic Severity of Mitral Valve Prolapse

**DOI:** 10.3390/life13020302

**Published:** 2023-01-21

**Authors:** Ming-Hsuan Huang, Sheng-Hui Tuan, Yun-Jeng Tsai, Wei-Chun Huang, Ta-Cheng Huang, Shin-Tsu Chang, Ko-Long Lin

**Affiliations:** 1Department of Physical Medicine and Rehabilitation, Kaohsiung Veterans General Hospital, No. 386, Ta-Chung 1st Road, Kaohsiung 81362, Taiwan; 2Department of Rehabilitation Medicine, Cishan Hospital, Ministry of Health and Welfare, Kaohsiung 842, Taiwan; 3Institute of Allied Health Sciences, College of Medicine, National Cheng Kung University, Tainan 701, Taiwan; 4Jia Huan Rehabilitation Clinic, Kaohsiung 830, Taiwan; 5Department of Critical Care Medicine and Cardiology Center, Kaohsiung Veterans General Hospital, Kaohsiung 813, Taiwan; 6School of Medicine, College of Medicine, National Yang Ming Chiao Tung University, Taipei 112, Taiwan; 7Department of Pediatrics, Kaohsiung Veterans General Hospital, Kaohsiung 813, Taiwan; 8School of Medicine, National Defense Medical Center, Taipei 11490, Taiwan; 9School of Medicine, College of Medicine, Kaohsiung Medical University, Kaohsiung 807, Taiwan; 10Department of Post-Baccalaureate Medicine, National Sun Yat-sen University, Kaohsiung 804, Taiwan

**Keywords:** floppy mitral valve, valvular heart disease, exercise capacity, cardiopulmonary assessment, left ventricular function

## Abstract

Patients with mitral valve prolapse (MVP) have been reported to have exercise intolerance. However, the underlying pathophysiological mechanisms and their physical fitness remain unclear. We aimed to determine the exercise capacity of patients with MVP through the cardiopulmonary exercise test (CPET). We retrospectively collected the data of 45 patients with a diagnosis of MVP. Their CPET and echocardiogram results were compared with 76 healthy individuals as primary outcomes. No significant differences regarding the patient’s baseline characteristics and echocardiographic data were found between the two groups, except for the lower body mass index (BMI) of the MVP group. Patients in the MVP group demonstrated a similar peak metabolic equivalent (MET), but a significantly lower peak rate pressure product (PRPP) (*p* = 0.048). Patients with MVP possessed similar exercise capacity to healthy individuals. The reduced PRPP may indicate compromised coronary perfusion and subtle left ventricular function impairment.

## 1. Introduction

Mitral valve prolapse (MVP) is a common valvular disorder that affects 2–3% of the general population [1], and the prevalence has been estimated at 2.2% in the Asian population [2]. Mitral valve prolapse is often considered asymptomatic. However, devastating outcomes could occur with disease progression, such as mitral regurgitation (MR), endocarditis, sudden death, and stroke [3,4,5,6,7,8]. MVP represents the most common cause of primary MR in the Western world [9]. Although a high mortality rate is associated with moderate-to-severe MR and a reduced left ventricular ejection fraction [10], sudden deaths caused by ventricular arrhythmia have also been reported [5,11]. Even though MVP is the most common valvular abnormality in industrialized countries [12], some patients do not seek medical help until complications occur. Freed et al. even identified a proportion of MVP patients in a community-based setting whose clinical profile was more benign than previously indicated by the literature [13]. Such patients may need long-term follow-up and observation for possible complications; however, less is known about the determination of the prognosis. Considering the multidimensional nature of this disease, risk stratification should carefully incorporate markers of left ventricular dysfunction, arrhythmic burden, and exercise capacity.

Some MVP patients with exercise intolerance have been reported. These patients either present with symptoms that limit their exercise performance at school or feel more exhausted during physical activity than their peers. However, through exhaustive clinical and laboratory investigations, the symptoms were often disproportionate to objective findings. According to the literature, reduced left ventricular filling [14], autonomic dysfunction [15,16], a hyperadrenergic state [16], catecholamine excess [14], metabolic disturbances, or combinations of the above-mentioned factors are all possible explanations for the constellation of symptoms [17,18,19]. Nevertheless, the pathophysiological mechanisms of most of the associations remain elusive.

Most methods used to evaluate left ventricular function in cardiac diseases are performed with the patient at rest. As symptoms are frequently related to physical exercise, they often poorly correlate with left ventricular function indexes obtained at rest [20]. The cardiopulmonary exercise test (CPET) is a useful tool in assessing functional capacity during exercise and risk evaluation, as well as determining the prognosis. Also, the CPET can identify factors that limit physiologic variables [20]. To the best of our knowledge, the use of this method in patients with MVP has not been reported in the literature. Obtaining measurements of exercise responses integrated by cellular, cardiovascular, and ventilatory systems is important to understand their exercise abilities, predict prognosis, and promote following exercise training programs. Thus, we aimed to determine whether the patients with MVP had comparable physical fitness, measured by the CPET, to normal subjects.

## 2. Material and Methods

### 2.1. Participants and Study Design

This was a retrospective observational cohort study. We reviewed the medical records and analyzed data collected at a single medical center in southern Taiwan from April 2012 to May 2021. During this period, patients with various symptoms, such as palpitations, atypical chest pain, dyspnea, or syncope, and eventually diagnosed by cardiologists with MVP via the two-dimensional echocardiographic study were enrolled. The inclusion criteria were as follows: patients with a diagnosis of MVP who completed both transthoracic echocardiography and CPET. Patients with other congenital heart diseases (e.g., patent ductus arteriosus, ventricular septal defect, pulmonary artery stenosis, atrial septal defect), preexisting pulmonary diseases, significant coronary artery diseases, and also patients with missing data or incomplete exercise tests were excluded. To compare their exercise capacity, age- and sex-matched healthy controls were selected from our database.

All patients underwent body composition measurement followed by a pulmonary function test and a symptom-limited treadmill exercise test. Patients’ characteristics and demographic variables consisted of age, gender, body weight, body height, BMI, body fat, blood pressure, pulse rate during rest, and the echocardiogram data generated by experienced cardiologists. This study was approved by the Institutional Review Board of the Kaohsiung Veterans General Hospital (VGHKS 17-CT11-11). All study procedures were performed per the principles of the Helsinki Declaration.

### 2.2. Cardiopulmonary Exercise Testing

Exercise capacity was evaluated via a symptom-limited CPET consisting of a treadmill, a gas analyzer, a flow module, and an electrocardiography monitor (Metamax 3B; Cortex Biophysik Co., Leipzig, Germany). Informed verbal consent and written consent were obtained from the subjects and their families, respectively. All subjects and their families received detailed explanations before the exercise test and fully understood the protocol and purpose of this test. We measured oxygen consumption at peak exercise (peak VO_2_) and anaerobic threshold (AT VO_2_) per the Bruce ramp protocol, as suggested by the American College of Sports Medicine. The peak VO_2_ and carbon dioxide production were measured via the breath-by-breath method. The metabolic equivalent (MET), which is equal to 3.5 milliliters of oxygen per kilogram of body mass per minute, was calculated after measuring VO_2_. The peak MET and AT MET were obtained as the maximal value throughout the whole exercise test and the value at the anaerobic threshold, respectively. The percentage of the peak VO_2_ to the predicted value (peak VO_2_%) was the percentage compared with normal standards of cardiopulmonary responses to exercise in Taiwan [21]. The heart rate (HR), blood pressure (BP), and minute ventilation (VE) were also recorded. The carbon dioxide production divided by oxygen consumption was calculated as the respiratory gas exchange ratio (RER). The anaerobic threshold was decided by the ventilatory efficiency (i.e., the VE/VCO_2_ slope) and ventilatory equivalents for the oxygen ratio (i.e., the VE/VO_2_ slope) production methods. The oxygen pulse was defined as the ratio of oxygen consumption to heart rate (i.e., VO_2_/HR), namely the volume of oxygen ejected from the ventricles with each cardiac contraction. The oxygen uptake efficiency slope (OUES) was calculated as the slope of the curve representing the logarithmic relationship between ventilation and oxygen consumption [22]. During the test, the rate pressure product, the product of the HR and the systolic arterial pressure, were calculated, and the maximum value was acquired as the peak rate pressure product (PRPP) [23].

The test was terminated when the maximum effort was obtained or the subjects could no longer continue due to physical symptoms. The maximum effort was achieved when RER > 1.1, the peak HR > 200 beats per minute, or the HR was >85% of the age-predicted maximum. All tests were performed smoothly under the supervision of well-trained physiatrists.

### 2.3. Echocardiography

All patients underwent two-dimensional echocardiographic studies with both color flow and spectral Doppler examinations using a transducer probe with a frequency of more than 5 MHz. The diagnosis of MVP was based on the criterion of mitral valve displacements of more than 2 mm above the mitral annulus in the long-axis view. The disease spectrum of MVP ranges from no or unremarkable leakage to severe MR. According to Kim et al., the complication rate was significantly higher in MVP patients who showed a progression in the severity of MR [24]. The degree of MR was graded using integrative assessments, as recommended by the American Society of Echocardiography [25]. The parameters assessed included the mitral jet area as a percentage of the left atrial area, the vena contracta width, and the calculation of the effective regurgitant orifice area (EROA) using the proximal isovelocity surface area (PISA) method. The final determination of MR into the following grades was made by the cardiologists based on his or her expertise and experience: mild (1+/4+), moderate (2+/4+), moderate-to-severe (3+/4+), and severe (4+/4+). To investigate whether patients with moderate or greater MR had a relatively low level of aerobic fitness, patients with MVP were divided into two groups, according to the degree of MR for the subgroup analysis, as follows: the mild or no MR group (MR < 2+) and the moderate or greater MR group (MR ≥ 2+).

Apart from diagnosing MVP, routine examinations of cardiac structures, such as ventricular sizes or wall motion abnormalities, were also screened. Following the guidelines and standards for the performance of echocardiography by the American Society of Echocardiography [26], variables, such as the left atrium diameter (LAD), left ventricular shortening fraction, left ventricular ejection fraction (LVEF), end-diastolic left ventricular internal diameter (LVID), and end-systolic LVID, were measured.

## 3. Statistical Analysis

All statistical analyses were performed using SPSS version 22.0 (IBM Corp., Armonk, NY, USA). Continuous data are expressed as mean ± SD, and categorical data are expressed as frequencies and percentages.

Data were tested for normality and homoscedasticity before each analysis. Analyses were conducted using the independent t-test and Pearson’s chi-square test, as appropriate. Fisher’s exact test was used for between-group comparisons of MR severity. The primary outcome of this study was the peak MET, measured by submaximal exercise testing using the CPET. We used independent t-tests for comparisons between normally distributed variables and Mann–Whitney U tests for comparisons between non-normally distributed variables. A *p*-value of ≤0.05 was considered statistically significant.

## 4. Results

### 4.1. Demographics

Of the 59 MVP patients who met the eligibility criteria, there were 5 patients with cardiac structural diseases and 9 patients with missing data. Therefore, 45 patients were included. Being selected from our database of healthy controls, 76 patients with age and sex matches and having no significant structural abnormality, contractile dysfunction, altered wall motion, or abnormal left ventricular mass constituted the control group.

The mean ages of patients with MVP and those in the control group were 18.78 ± 8.73 and 16.61 ± 8.94 years, respectively. Table 1 shows the demographic characteristics of the participants in the two groups. There was no statistically significant difference in age, sex, body height, body weight, body fat, resting BP, or resting HR. The patients in the MVP group had a significantly lower mean BMI (19.72 ± 2.99) than those in the control group (21.21 ± 4.54) (*p* = 0.032).

### 4.2. Echocardiographic Data and Pulmonary Function Tests

The echocardiographic variables, including the LVIDs, LAD, left ventricular shortening fraction, and LVEF, did not differ significantly between the MVP group and the control group (Table 2). Among the patients in the MVP group, 13 (28.9%), 13 (28.9%), 18 (40%), and 1 (2.2%) had no, mild, moderate, and moderate-to-severe MR, respectively. There were no patients with severe MR. In contrast, most patients (92.1%) in the control group had no MR. MR severity differed significantly between the two groups (*p* < 0.001). There were no significant differences in any of the routine spirometry measurements, including forced vital capacity (FVC), forced expiratory volume in 1 s (FEV1), and maximal voluntary ventilation, between the MVP group and the control group, except for the FEV1/FVC. The MVP group had a lower mean FEV1/FVC (83.38% ± 8.83%) than the normal group (87.24% ± 9.42%) (*p* < 0.05). None of the participants had a value below 70%; thus, no obstructive lung disease was diagnosed.

### 4.3. Cardiopulmonary Exercise Testing

Table 3 presents the comparison of CPET variables between the two groups. No statistically significant differences were observed in the routine parameters measured during the standard exercise test, including peak MET, AT MET, peak VO2%, peak HR, peak diastolic and systolic BP, VE, and peak RER. The OUES, PRPP, VE/VCO_2_, and oxygen pulse were also recorded. There were no significant differences between the two groups except for the PRPP, where the MVP group presented with a lower value (27277 vs. 29078, *p* = 0.048).

CPET parameters were investigated for MVP patients according to their degree of mitral regurgitation. These results are also available in Table 3. There were no statistical differences between outcomes for patients with mild or no MR (grade < 2+, n = 26) or moderate or greater MR (grade ≥ 2+, n = 19).

## 5. Discussion

Our study shows that patients diagnosed with MVP mostly have normal exercise capacity compared with normal individuals of the same sex and similar age, body height, and body weight. However, we found significantly lower PRPP in patients with MVP.

Despite the prevalence of mitral valve prolapse and the known exercise intolerance, there was little literature about the exercise capacity of MVP patients. Our study reports normal exercise capacity in both groups. A similar finding was reported in a study that included an analogous population but without elaborating on the details of the exercise test results and focusing on the arrhythmia burden of the patient group instead [27]. Some other studies have reported abnormal exercise tests by recognizing ST-segment depression [28,29] or by total exercise duration and maximum workload [14] instead of comparing the VO_2_. Interestingly, a study by Koutlianos et al., in which soccer players were enrolled, conveyed the concept that athletes with mild or moderate MVP presented with comparable maximum VO_2_ to those without MVP; furthermore, both groups possessed better results than the group of normal controls [30]. Neither of these studies emphasized the precise exercise capacity demonstrated by the CPET results.

There is some evidence of an association between mitral valve prolapse and slender body habitus in adults, teenagers, and pediatric populations [4,13,31,32,33,34]. Indeed, our study demonstrates significantly lower BMIs in patients with MVP. To elaborate on the influence of BMI on VO_2_, previous studies have yielded conflicting results. Horwich et al. concluded that, among patients with chronic heart failure, elevated BMI is not only associated with higher absolute peak VO_2_ (L/min), as expected, but is also an independent predictor of lower relative peak VO_2_ (mL/kg/min) [35]. Conversely, among patients with unexplained dyspnea, Antranik et al. found no significant correlation between BMI and VO_2_ (L/min) and stated that BMI did not impact the diagnostic yield of CPET [36]. Although the study population is different, body fat, which consumes almost no oxygen, might somehow contribute to the relationship between obesity and lower peak VO_2_. As noted above, the relevance between peak VO_2_ and BMI is more prominent in the overweight population, and the peak VO_2_ lean (peak VO_2_ for lean body mass) might be more predictive [37]. As our study participants were all slender, regardless of their groups, differences in BMI had little influence on the peak VO_2_.

Given that CPET results represent the integrated function of the cellular, cardiovascular, and ventilatory systems, and the oxygen transport and utilization depend on both cardiovascular and pulmonary responses, the lung function test provides important clues when discussing exercise tolerance. Despite the difference in FEV_1_/FVC between the MVP group and the control group found in our study, both values were within normal limits, which is compatible with a previous study [38]. Much literature has studied the association between lung function tests and exercise capacity in different ways. Pinoto-Plata et al. demonstrated that as COPD progressed, there was a significant reduction in exercise capacity at every stage except for stage I (i.e., FEV1 > 80% predicted), who had comparable CPET results to normal subjects [39]. Carter et al. mentioned similar results, i.e., that exercise capacity in a portion of patients did not change until the disease was at moderate or severe levels [40]. Ostolin et al. reported that moderate-to-vigorous physical activity is positively correlated to FVC and FEV1 in the general population. Ostolin et al. reported that moderate-to-vigorous physical ac- tivity is positively correlated to FVC and FEV1 in general population. However, these re lationships became non-significant when controlled by CPET [41] evidence that exercise intolerance among this population was associated with lower FEV1/FVC.

The PRPP gives an accurate reflection of the myocardial oxygen demand and myocardial workload [42,43]. To investigate the influence of coronary perfusion on PRPP, Tuan et al. evaluated the exercise capacity of patients with Kawasaki disease in different coronary artery z-scores. Patients with no coronary involvement showed higher PRPP than those who had coronary artery dilation [44]. Santos et al. also concluded that when maladaptive eccentric left ventricular hypertrophy occurs, the myocardial oxygen uptake increases, leading to the exhaustion of the coronary blood flow reserve [45].

The ability to reach higher PRPP is associated with more adequate coronary perfusion; that is, the low value of PRPP suggests a significant coronary perfusion compromise and decreased left ventricular function [46]. It is well-established that some MVPs could progress into MR or even left ventricular overload and dysfunction when approaching advanced stages [47,48,49]. The LVEF is often overestimated in the presence of valvular diseases because it simply reflects the entrance and exit of blood in the left ventricle [50]. Therefore, the LVEF may remain in the normal or supernormal range for long periods, even if alterations in contractility occur. In one study comparing MVP patients with a preserved LVEF and control participants using cardiovascular magnetic resonance, left ventricle enlargement, basal inferolateral hypertrophy, higher posterior annular excursion, and reduced shortening of the papillary muscles differed significantly, and the difference remained significant in patients without evident MR [51]. We found no significant difference in the LVEF between the two groups; however, the altered geometry and mechanics of the mitral annulus may account for a repetitive mechanical stretch to the valve and left ventricular myocardium, as mentioned above [50]. Meanwhile, a higher myocardial work rate associated with repeated traction increases the energy demand and oxidative stress in this region, which can eventually provoke localized hypertrophy and fibrosis [1]. To detect left ventricular function abnormalities early, we assumed that PRPP could be a potential indicator in MVP patients; however, further prospective studies on the association between PRPP and left ventricular-related parameters are needed to confirm its predictive value.

Boudoulas et al. reported that there are two groups of MVP patients. In one group of patients, symptoms, physical findings, and natural history are directly related to progressive MR and its complications through their effects on the left ventricular structure. In the other group of patients, symptoms could not be explained by the degree of MR alone; neuroendocrine dysfunction has been implicated in the explanation of the symptoms noticed in this group. The phenomenon observed among patients in this group is referred to as mitral valve prolapse syndrome (MVPS) [52]. Although most MVP patients have an unimpaired functional capacity, some patients with uncomplicated prolapse may have decreased exercise tolerance in formal testing [14,53]. There is little information about the mechanism underlying exercise intolerance, and the most mentioned descriptions are correlated with MVPS. Patients with MVPS present with a symptom complex that results from various forms of neuroendocrine or autonomic dysfunction, including palpitations, orthostatic phenomena, syncope, presyncope, exercise intolerance, dyspnea, and neuropsychiatric symptoms [15,19]. In MVPS, alterations of the heart, kidney, adrenals, and autonomic nervous system coexist and interact, creating a complex “neuro-endocrine cardiovascular process” that could account for many of the symptoms otherwise unexplained based on the valvular abnormality alone [19].

The quantification of the exercise capacity of patients with MVP is important for several reasons. The disease spectrum could range from asymptomatic to severe regurgitation. Nevertheless, there is no evident association between disease severity and physical activity. Through participation in CPET evaluations, self-efficacy is established by performing comparable physical fitness. Further, exercise limitations due to anxiety alone could be distinguished from organic disease. Most importantly, due to various symptoms and impaired quality of life in these MVP patients (especially those with MVPS), interventions with exercise training based on CPET reports would be reasonable. The physiological and psychological benefits of exercise training in MVP patients have been documented and increasingly recognized [17,30,54]. Although it is difficult to determine to what extent exercise intolerance is dependent on MVPS, it is evident that, through programmed exercise training, improvements in symptoms and quality of life were observed in the symptomatic patient subset.

Our study had certain limitations. First, this study was conducted in a single medical center in southern Taiwan; therefore, the results may not be applicable to the general population. Due to its small sample size and retrospective nature, further large-sample, prospective studies are needed to better understand the interaction between left ventricular function and exercise capacity in these populations. In addition, those who had severe MR and serious complications of prolapse were not included in our study. Indeed, speaking of the extent of regurgitation, our patients are mostly those with mild and moderate MR. Patients with prominent symptoms may directly seek surgical help instead of undergoing detailed examinations, which contributed to selection bias. Thus, it is possible that the results may not be generalizable to patient subgroups that present with severe MR or unbearable symptoms with exertion. Although stratification by the severity of MR has revealed no statistical differences between the two groups, further large-sample studies including more patients with mild-to-moderate and even severe MR are needed. Also, echocardiographic details, such as the left atrium volume, diastolic function with the E/A ratio, leaflet thickness, mitral annular diameters, and global longitudinal strain, are lacking in this study. Meanwhile, the limitations in assessing MR severity should be noted when interpreting the results of this study. For instance, the PISA method can cause an underestimation of the severity and inaccurate measurements of the eccentric jet [55]. Future studies should include three-dimensional echo imaging or directly measure regurgitation volume using cardiac magnetic resonance for better quantification of the severity of regurgitation [56]. Last but not least, a combination of exercise echocardiography with CPET is warranted in future studies to obtain data for both the physiological variables and the visible ventricle response with the contractile reserve in MVP patients during exercise.

## 6. Conclusions

In conclusion, this study reports generally comparable physical fitness between MVP patients and their healthy peers, including AT MET, peak MET, VE/VCO_2_, and OUES. Although we found significantly decreased PRPP in the MVP group, its predictive value should be interpreted with caution. To further understand the complex interaction between the anatomical, pathophysiological, and hemodynamic aspects of MVP patients, more investigations are needed.

## Figures and Tables

**Table 1 life-13-00302-t001:** Demographics and baseline characteristics.

Characteristics	MVP Group, Mean ± SD (n = 45)	Control Group, Mean ± SD (n = 76)	*p*-Value
Age, year	18.8 ± 8.7	16.6 ± 8.9	0.195
Sex, male/female	8/37	25/51	0.071
Height, cm	161.8 ± 8.1	161.1 ± 11.4	0.748
Weight, kg	51.9 ± 9.9	55.8 ± 15.3	0.099
BMI, kg/m^2^	19.7 ± 3.0	21.2 ± 4.5	0.032
Body fat, %	21.7 ± 6.7	22.2 ± 8.5	0.721
Resting SBP, mmHg	117.1 ± 13.2	120.7 ± 16.4	0.188
Resting DBP, mmHg	70.3 ± 8.3	71.5 ± 10.3	0.498
Resting HR, bpm	88.1 ± 14.5	85.9 ± 17.7	0.482
Hemoglobin, g/dL	13.2 ± 1.4	13.6 ± 1.3	0.147

BMI, body mass index; DBP, diastolic blood pressure; HR, heart rate; SBP, systolic blood pressure.

**Table 2 life-13-00302-t002:** Echocardiographic data and lung function.

Test Result	MVP Group, Mean ± SD (n = 45)	Control Group, Mean ± SD (n = 76)	*p*-Value
Echocardiographic data			
LVIDs, cm	2.5 ± 0.4	2.6 ± 0.4	0.427
LVIDd, cm	4.1± 0.5	4.2± 0.5	0.502
LAD, cm	2.2 ± 0.4	2.2 ± 0.5	0.802
LV shortening fraction, %	39.2 ± 6.2	38.0 ± 8.5	0.408
LVEF, %	69.3 ± 7.7	69.2 ± 6,1	0.964
The presence of MR, n (%)			<0.001
No (0)	13 (28.9%)	70 (92.1%)	
Mild (1+)	13 (28.9%)	4 (5.3%)	
Moderate (2+)	18 (40%)	2 (2.6%)	
Moderate-to-severe (3+)	1 (2.2%)	0 (0%)	
Lung function test			
FVC, L	2.77 ± 0.72	2.88 ± 0.84	0.456
FEV1, L	2.33 ± 0.66	2.51 ± 0.77	0.176
MVV, L	65.79 ± 21.45	71.99 ± 27.19	0.197
FEV1/FVC, %	83.38 ± 8.83	87.24 ± 9.42	0.027

FEV1, forced expiratory volume in the first second; FVC, forced vital capacity; LAD, left atrium diameter; LV, left ventricle; LVEF, left ventricular ejection fraction; LVIDd, left ventricular internal dimension in diastole; LVIDs, left ventricular internal dimension in systole; MR, mitral regurgitation; MVV, maximum voluntary ventilation.

**Table 3 life-13-00302-t003:** Comparisons of cardiopulmonary exercise test variables between patients with mitral valve prolapse (MVP) and control participants (left column) and between MVP patients < moderate vs. ≥moderate mitral regurgitation (right column).

Test Result	MVP Group (n = 45) Mean ± SD	Control Group (n = 76) Mean ± SD	*p*-Value	MVP with MR < 2+ * (n = 26) Mean ± SD	MVP with MR ≥ 2+ * (n = 19) Mean ± SD	*p*-Value
AT MET	5.8 ± 1.2	6.6 ± 1.3	0.093	5.8 ± 1.3	5.8 ± 1.1	0.883
Peak MET	9.0 ± 1.7	9.5 ± 1.8	0.108	9.0 ± 1.8	9.0 ± 1.6	0.888
PP VO2% (%)	84.8 ± 16.3	82.1 ± 16.4	0.384	81.3 ± 13.1	89.7 ± 19.1	0.109
AT HR (bpm)	139.9 ± 13.4	141.2 ± 14.3	0.606	139.9 ± 15.7	139.8 ± 9.7	0.991
Peak HR (bpm)	175.4 ± 12.3	177.5 ± 12.1	0.376	175.5 ± 12.4	175.3 ± 12.4	0.961
Peak HRR (bpm)	26.0 ± 8.8	27.2 ± 10.0	0.514	25.1 ± 7.5	27.3 ± 10.4	0.425
Peak SBP (mmHg)	155.7 ± 24.0	164.2 ± 30.5	0.090	155.8 ± 21.5	155.5 ± 27.6	0.970
Peak DBP (mmHg)	72.5 ± 14.6	75.3 ± 16.3	0.331	69.0 ± 13.7	77.2 ± 14.9	0.065
Peak VE (liters)	50.5 ± 13.0	54.4 ± 15.2	0.152	52.1 ± 13.1	48.3 ± 12.8	0.336
Peak RER	1.2 ± 0.1	1.1 ± 0.6	0.141	1.2 ± 0.1	1.2 ± 0.1	0.677
OUES	1.8 ± 0.4	2.3 ± 3.3	0.243	1.8 ± 0.4	1.7 ± 0.4	0.762
VE/VCO2	27.0 ± 3.9	26.7 ± 5.6	0.814	26.7 ± 3.4	27.3 ± 4.5	0.638
PRPP	27277 ± 4374	29078 ± 5387	0.048	27389.3 ± 4568.8	27124.5 ± 4211.7	0.844
Oxygen pulse	9.3 ± 2.2	9.9 ± 2.2	0.129	9.3 ± 2.3	9.2 ± 2.2	0.906

AT VO2, oxygen consumption when anaerobic threshold; HRR, heart rate reserve; MET, metabolic equivalent; OUES, oxygen uptake efficiency slope; PP VO2%, age predicted peak oxygen consumption; PRPP, peak rate pressure product; RER, respiratory exchange ratio; VE, minute ventilation; VE/VCO2, ventilatory efficiency. * MVP with MR < 2+: MAP patients without MR or with mild MR noted by echocardiography; MVP with MR ≥ 2+: MVP patients with moderate or greater degrees of MR noted by echocardiography.

## Data Availability

The data that support the findings of this study are available from the corresponding author upon reasonable request.

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
