# Peer review of "Comparison of the Results of Cardiopulmonary Exercise Testing between Healthy Peers and Pediatric Patients with Different Echocardiographic Severity of Mitral Valve Prolapse"

_life, 2023, doi:10.3390/life13020302_

Round 1
Reviewer 1 Report
This study is about the comparison between MVP and control groups. The peak rate pressure product (PRPP) showed a difference between groups but did not found between different MR grades. RPP varies with exercise. PRPP gives an accurate reflection of the myocardial oxygen demand and myocardial workload. The higher the PRPP, the more will be myocardial oxygen consumption (MVO2) is associated with more adequate coronary perfusion. LV performed was similar between the two groups. The difference was found in lung function (FEV1/FVC). Therefore, the difference is related to lung or exercise loading. The author clearly mentioned the limitation is the limited number of this study.
Reviewer 2 Report
In this study authors evaluated the exercise capacity of patients with mitral prolaxe. They compared anthropometric, echocardiography and cardiopulmonary test data of 45 patients with mitral prolapse with those of 76 healthy subjects. They found three main differences between these two groups: a lower body mass index in the mitral prolapse group; an higher prevalence of mitral regurgitation in the prolapse group (as was to be expected); a lower peak rate pressure product again in the prolapse group. The paper is interesting and overall it appers to be well written
Authors hypothesize that the lower PRPP might be a marker of subclinical left ventricular dysfunction in patients with mitral prolapse. It would be interesting to know (if data are available) what the GLS values were in this group of patients since they could reflect changes in myocardial contranctility better than LVEF.
Moreover the exact meaning of the reduced PRPP is unclear. It would have been worth acquiring ecocardiography data at peak exercise in order to rule out left ventricular dysfunction during exercise.
Since the study is specifically designed for evaluating mechanisms of exercise intolerance the lack of echocardiography data during exercise constitutes a limitation and should be reported in the discussion paragraph
Was the level of exercise training comparable between the two groups? This could explains the observed differences in PRPP .
Pulse oxygen and VO2/workload relationship should be reported among CPX variables since they arre related with stroke volume
In this study authors evaluated the exercise capacity of patients with mitral prolaxe. They compared anthropometric, echocardiography and cardiopulmonary test data of 45 patients with mitral prolapse with those of 76 healthy subjects. They found three main differences between these two groups: a lower body mass index in the mitral prolapse group; an higher prevalence of mitral regurgitation in the prolapse group (as was to be expected); a lower peak rate pressure product again in the prolapse group. The paper is interesting and overall it appers to be well written
Authors hypothesize that the lower PRPP might be a marker of subclinical left ventricular dysfunction in patients with mitral prolapse. It would be interesting to know (if data are available) what the GLS values were in this group of patients since they could reflect changes in myocardial contranctility better than LVEF.
Moreover the exact meaning of the reduced PRPP is unclear. It would have been worth acquiring ecocardiography data at peak exercise in order to rule out left ventricular dysfunction during exercise.
Since the study is specifically designed for evaluating mechanisms of exercise intolerance the lack of echocardiography data during exercise constitutes a limitation and should be reported in the discussion paragraph
Was the level of exercise training comparable between the two groups? This could explains the observed differences in PRPP .
Pulse oxygen and VO2/workload relationship should be reported among CPX variables since they arre related with stroke volume
Reviewer 3 Report
The authors aimed to determine the differences between the physical fitness of patients diagnosed with MVP and normal subjects through CPET. They concluded that this study reports generally comparable physical fitness between MVP patients and their healthy peers, including AT MET, peak MET, VE/VCO2, and OUES. In addition, significantly decreased PRPP in the MVP group was observed.
General comments
This is a manuscript addressing a topic “Comparison of the Results of Cardiopulmonary Exercise Testing between Healthy Peers and Pediatric Patients with Different Echocardiographic Severity of Mitral Valve Prolapse”. Some concerns need to be addressed.
Specific comments
Major concerns
1) Line 70: To the best of our knowledge, the use of this method in patients with MVP has not been reported in the literature. The clinical gap would be relatively vague. As the authors mentioned in discussion (Line 281), more reasons would be necessary in the introduction.
2) Line 130: more detail of methods for MR grading would be necessary (PISA, volumetric, or others). Because the most critical point in this paper is the MR grading, the method would be important. Because of the eccentric MR jet, the quantification of MR is sometimes difficult.
Minor concerns
1) Line 150: the primary outcomes are vague. What was the primary hypothesis of the authors?
2) Table 1-3: abbreviations in the footnote would be alphabetical order. Mean±SD and number of the patients are missing.
Round 2
Reviewer 1 Report
The author clearly responds to the comments.
Author Response
Respond to reviewer 1
Thank you very much for making detailed review and valuable comments. We deeply appreciate your help on this manuscript.